# Systematic over-expression of secondary metabolism transcription factors to reveal the pharmaceutical potential of *Aspergillus nidulans*

Shuhui Guo [1,2] ✉, Lakhansing Pardeshi[3], Longguang Qin[1], Chris Y. Cheung[4], Xiaofeng Liu[1], Lu Fan[1], Chi Cheng Mok[1], Chirag Parsania [1], Zhiqiang Dong[1], Ben C. B. Ko[4], Kaeling Tan[1,3] & Koon Ho Wong [1,5,6] ✉

Many life-saving drugs are derived from fungal secondary metabolites, and the rich diversity of these metabolites is a gold mine of bioactive compounds for drug discovery. However, the biosynthetic genes for most secondary metabolites remain transcriptionally silent in fungi, posing a significant bottleneck in their discovery. Here, we apply a systematic approach to separately over-express 51 secondary metabolism-related transcription factors using the strong inducible promoter of the *xylP* gene from *Penicillium chrysogenum*. Growing the individual secondary metabolism transcription factor over-expression strains under inducible conditions leads to the production of a collection of diverse metabolites with anti-bacterial, anti-fungal and anti-cancer activities. The overall approach and the over-expression system established in this study are broadly-applicable, providing a valuable means to revealing the pharmaceutical potentials of fungi.

Fungal secondary metabolites represent a rich repertoire of natural products[1,2]. Although numerous secondary metabolite biosynthetic gene clusters (BGCs) have been identified bioinformatically in thousands of fungal genomes, the functionality of these predicted BGCs and their potential to produce metabolites awaits validation. More importantly, most BGCs remain transcriptionally silent under standard laboratory conditions. The transcriptional silencing is believed to be due to epigenetic regulation and/or regulation of BGC-specific transcription factors (TFs)[2–4], which are defined merely based on the physical location of their genes within a BGC. Consequently, the vast majority of fungal secondary metabolites have not been identified, and their bioactivity remains unknown. Despite the development of various methods and significant efforts to explore this rich resource[2–11], most studies have been limited to a low-throughput approach—activating one cluster at a time to induce the production of its secondary metabolite, which is then tested for bioactivity[10,12–14]. Notably, a recent study has systematically over-expressed 58 BGC-specific transcription factors of *Aspergillus niger*, leading to the production of potentially novel compounds[15].

Although the bioactivity of those compounds was not determined in that study, such systematic approach provides a valuable means to evaluate the pharmaceutical potential of a given fungus for drug discovery.

In the well-studied filamentous fungus *Aspergillus nidulans*, 71 secondary metabolite BGCs had been identified from its genome sequence bioinformatically more than a decade ago[16]. However, to date, more than half of them are still uncharacterized and their pharmaceutical potentials unknown. In a previous study, 33 BGC-specific TFs from 17 predicted secondary metabolite BGCs of *A. nidulans* were systematically over-expressed using the ethanol-inducible *alcA* promoter[17]. Contrary to previous successes with this over-expression (OE) strategy[18,19], only three secondary metabolite BGCs were activated to produce detectable levels of their compounds. This leaves an impression that the secondary metabolism (SM) TF OE strategy may not be universally applicable for all clusters. Consequently, the straightforward strategy has not been exploited for systematic screening of bioactive metabolites. Here, we systematically over-express 51 secondary metabolism related TFs in

[1]Faculty of Health Sciences, University of Macau, Macau SAR, China. [2]School of Health and Nursing, Wuxi Taihu University, Wuxi, China. [3]Genomics, Bioinformatics & Single Cell Analysis Core, Faculty of Health Sciences, University of Macau, Macau SAR, China. [4]Department of Applied Biology and Chemical Technology, The Hong Kong Polytechnic University, Hong Kong SAR, China. [5]Institute of Translational Medicine, Faculty of Health Sciences, University of Macau, Macau SAR, China. [6]MoE Frontiers Science Center for Precision Oncology, University of Macau, Macau SAR, China. ✉e-mail: guosh@wxu.edu.cn; koonhowong@um.edu.mo

*A. nidulans* using a strong inducible promoter and demonstrated the power of the systematic TF OE approach for drug discovery.

## Results and discussion

### Expression of SM TF from a strong promoter facilitates secondary metabolite BGC gene activation

Upon examining the experimental setup of the previous *A. nidulans* study[17] and comparing with the recent *A. niger* study[15], a potential explanation for the low success rate emerged. The *alcA* promoter used in the *A. nidulans* study[17] may not sufficiently elevate the transcriptional levels of most BGC-specific TF genes at their native genomic locus, which is believed to be regulated by repressive chromatin[20,21]. In addition, those TFs may be negatively controlled post-transcriptionally.

We hypothesized that most of secondary metabolite cluster genes would be induced if their cluster-specific TFs are sufficiently over-expressed (i.e. expressed at a level higher than that by the *alcA* promoter) and, therefore, undertook a similar systematic approach with slight modifications. First, we selected the *xylP* promoter from *Penicillium chrysogenum*, which is tunable and stronger than the *alcA* promoter and other commonly used promoters (e.g., *pcbC* and *niaD*), for conditional induction[22–24]. Second, we targeted the TF OE construct to the *yA* gene locus, which is supposedly not suppressed by repressive chromatin structures.

To test the feasibility of this strategy, the SM TF AN6790, which was over-expressed in the previous study[17] but failed to induce cluster activation (i.e., no novel product was detected in the *alcA*(p)::AN6790 strain after induction), and AflR (the best-studied cluster-specific activator of the biosynthetic genes for sterigmatocystin) were used as proof-of-principle examples. While both AN6790 and AflR proteins could not be detected under normal control conditions (i.e. without xylose), their levels were significantly induced in the presence of xylose (Fig. 1a), confirming their conditional expression.

To determine if the over-expression could successfully induce SM production and to compare with the previous studies[15,17], we used liquid culture conditions to analyze and compare metabolites production. The wild-type, AN6790-OE and AflR-OE strains were grown in ANM medium for 48 h at 37 °C, followed by addition of 1% xylose, and cultured for an additional three days. After five days of culture, the wild-type strain produced mauve pigment, while the AN6790-OE and AflR-OE strains produced orange and yellow pigments, respectively (Fig. 1b). Given that many fungal metabolites are pigmented[25], these color differences in the media of the different OE strains suggest the production of diverse metabolites by the OE strains. Indeed, LC-MS analysis of the growth media from the AflR-OE strain confirmed the production of sterigmatocystin (ST) (Fig. 1c). Furthermore, a unique metabolite profile was also observed for the AN6790-OE strain comparing to the wild type (WT) (Fig. 1d). Therefore, the revised OE

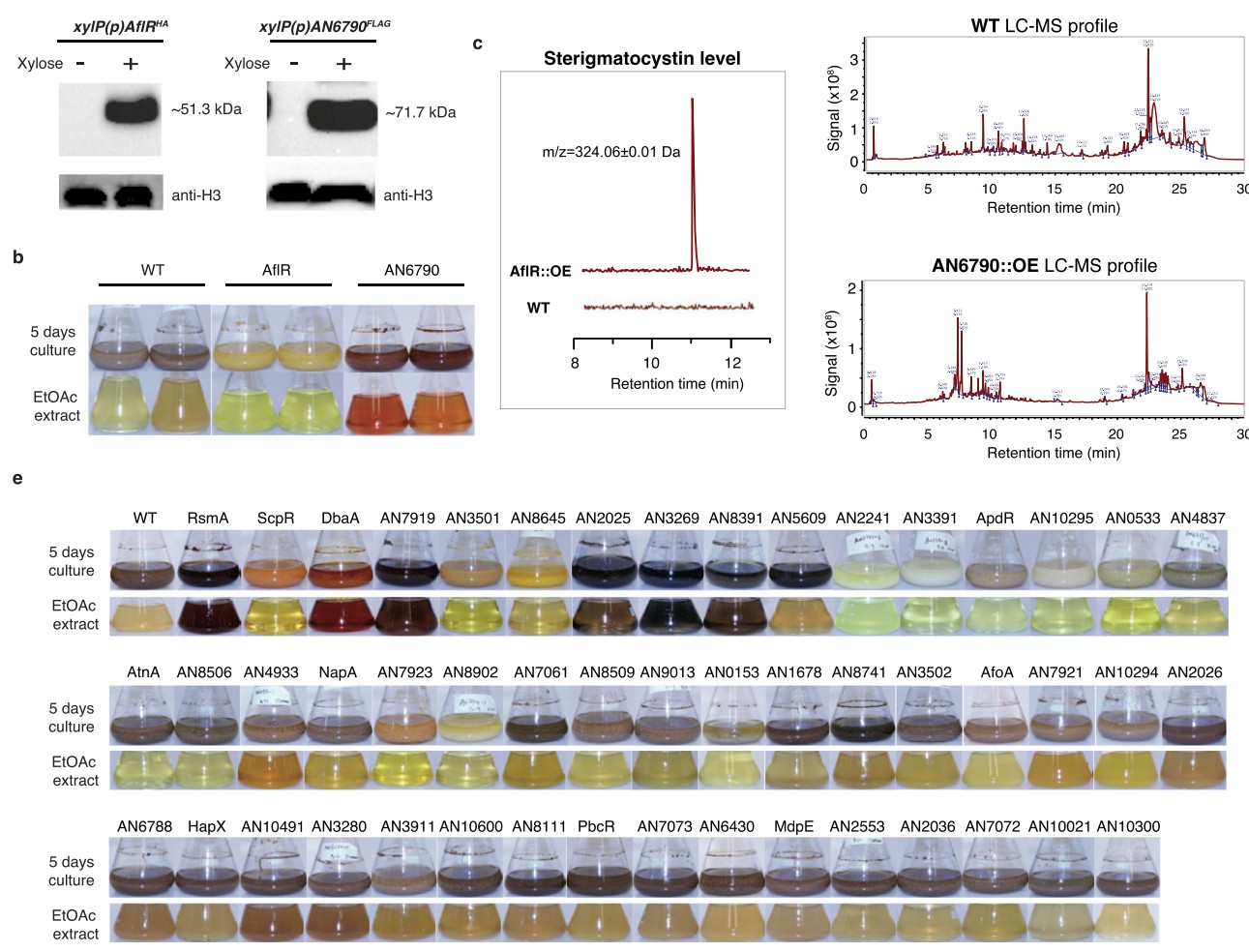

**Fig. 1 | Over-expression of SM TFs lead to the production of diverse metabolites.** **a** Western blot analysis of AflR[HA] (MW = 51.3 kD) and AN6790[FLAG] (MW = 71.7 kD) in the respective over-expression (OE) strains grown in the presence (+) and absence (-) of the xylose inducer. The expected protein size is indicated; **b** Images of fermentation broth (top panel) and their EtoAc extracts (bottom panel) of the indicated over-expression strains grown with 1% xylose induction; **c**, **d** LC-MS profiles of the EtoAc extract from the wild-type and AflR over-expression strains with 1% xylose induction; **e** Images of fermentation broth (top panel) and their ethyl acetate (EtOAc) extracts (bottom panel) of the indicated over-expression strains grown with 1% xylose induction.

**Table 1 | Summary of SM TF over-expression strains**

| No. | TF | Cluster | Type | Source | Ref |
|---|---|---|---|---|---|
| 1 | mdpE | Monodictyphenone (mdp) cluster | Zn(II)$_2$Cys$_6$ | Published | 47 |
| 2 | AN0153 | Monodictyphenone (mdp) cluster | Myb | SMURF | |
| 3 | AN0533 | pkdA cluster | Zn(II)$_2$Cys$_6$ | SMURF | |
| 4 | mdpA | Monodictyphenone (mdp) cluster | WHR | Published | 47 |
| 5 | afoA | Asperfuranone (afo) cluster | Zn(II)$_2$Cys$_6$ | Published | 48 |
| 6 | AN10294 | AN10289 cluster | Zn(II)$_2$Cys$_6$ | SMURF | |
| 7 | AN10295 | AN10289 cluster | bZIP | SMURF | |
| 8 | AN10300 | AN10297 cluster | Zn(II)$_2$Cys$_6$ | SMURF | |
| 9 | AN10491 | AN10486 cluster | Zn(II)$_2$Cys$_6$ | SMURF | |
| 10 | AN10600 | AN4827 cluster | Zn(II)$_2$Cys$_6$ | SMURF | |
| 11 | pbcR | AN1594 (PbcR) cluster | Zn(II)$_2$Cys$_6$ | Published | 49 |
| 12 | AN1678 | AN1680 cluster | Zn(II)$_2$Cys$_6$ | SMURF | |
| 13 | AN2025 | | Winged Helix Regulator | AspGD | |
| 14 | AN2026 | | uncertain | AspGD | |
| 15 | AN2036 | Pkh cluster | Zn(II)$_2$Cys$_6$ | SMURF | |
| 16 | AN2241 | | Zn(II)$_2$Cys$_6$ | AspGD | |
| 17 | AN2553 | Emericellamide (eas) cluster | Zn(II)$_2$Cys$_6$ | SMURF | |
| 18 | AN3269 | AN3273 cluster | Zn(II)$_2$Cys$_6$ | SMURF | |
| 19 | AN3280 | AN3273 cluster | Zn(II)$_2$Cys$_6$ | SMURF | |
| 20 | AN3391 | Microperfuranone cluster | C$_2$H$_2$ and Zn(II)$_2$Cys$_6$ | SMURF | |
| 21 | scpR | inp cluster | C$_2$H$_2$ | Published | 50 |
| 22 | AN3501 | inp cluster | Zn(II)$_2$Cys$_6$ | SMURF | |
| 23 | AN3502 | inp cluster | Zn(II)$_2$Cys$_6$ | SMURF | |
| 24 | AN3911 | AN10486 cluster | Zn(II)$_2$Cys$_6$ | SMURF | |
| 25 | rsmA | | bZIP | Published | 51 |
| 26 | AN4837 | AN4827 cluster | Zn(II)$_2$Cys$_6$ | SMURF | |
| 27 | AN4933 | | Zn(II)$_2$Cys$_6$ | AspGD | |
| 28 | AN5609 | AN5610 cluster | Zn(II)$_2$Cys$_6$ | SMURF | |
| 29 | AN6430 | AN6431 cluster | Zn(II)$_2$Cys$_6$ | SMURF | |
| 30 | AN6788 | xptA-containing cluster | Zn(II)$_2$Cys$_6$ | SMURF | |
| 31 | AN6790 | xptA-containing cluster | Zn(II)$_2$Cys$_6$ | SMURF | |
| 32 | AN7061 | pkg cluster | Zn(II)$_2$Cys$_6$ | SMURF | |
| 33 | AN7072 | pkg cluster | Zn(II)$_2$Cys$_6$ | SMURF | |
| 34 | AN7073 | pkg cluster | Zn(II)$_2$Cys$_6$ | SMURF | |
| 35 | napA | | bZIP | Published | 52 |
| 36 | aflR | Sterigmatocystin (stc) cluster | Zn(II)$_2$Cys$_6$ | Published | 53 |
| 37 | atnN | AN7884 cluster | Zn(II)$_2$Cys$_6$ | Published | 54 |
| 38 | dbaA | Dba and F9775 clusters hybrid 1 | Zn(II)$_2$Cys$_6$ | Published | 30 |
| 39 | AN7919 | Dba and F9775 clusters hybrid 2 | Zn(II)$_2$Cys$_6$ | SMURF | |
| 40 | AN7921 | Dba and F9775 clusters hybrid 2 | Zn(II)$_2$Cys$_6$ | SMURF | |
| 41 | AN7923 | Dba and F9775 clusters hybrid 2 | Zn(II)$_2$Cys$_6$ | SMURF | |
| 42 | AN8111 | AN8105 cluster | Zn(II)$_2$Cys$_6$ | SMURF | |
| 43 | hapX | AN8249 cluster | bZIP | Published | 55 |

**Table 1 (continued) | Summary of SM TF over-expression strains**

| No. | TF | Cluster | Type | Source | Ref |
|---|---|---|---|---|---|
| 44 | AN8391 | Austinol (aus) cluster | Zn(II)$_2$Cys$_6$ | SMURF | |
| 45 | apdR | Aspyridone (asp) cluster | Zn(II)$_2$Cys$_6$ | Published | 18 |
| 46 | AN8506 | Terriquinone (tdi) cluster | Zn(II)$_2$Cys$_6$ | SMURF | |
| 47 | AN8509 | Terriquinone (tdi) cluster | Zn(II)$_2$Cys$_6$ | SMURF | |
| 48 | AN8645 | | Zn(II)$_2$Cys$_6$ | AspGD | |
| 49 | mtfA | | C$_2$H$_2$ | Published | 56 |
| 50 | AN8902 | AN8910 cluster | Zn(II)$_2$Cys$_6$ | SMURF | |
| 51 | AN9013 | AN9005 cluster | Zn(II)$_2$Cys$_6$ | SMURF | |

setup can successfully activate SM gene expression and production from cryptic secondary metabolite BGCs.

## Systematic overexpression of transcription factors implicated in secondary metabolism leads to production of diverse metabolites

To further demonstrate the valuable potential of the strategy, we set out to systematically activate secondary metabolite BGCs in *A. nidulans* to screen for novel metabolites with pharmaceutical properties. TFs with potential roles in SM regulation were selected using several approaches. First, TFs located within predicted secondary metabolite BGCs annotated by SMURF (Secondary Metabolite Unknown Regions Finder) (http://www.jcvi.org/smurf/index.php)[26] or clusters identified in previous publications were considered candidates[16,27]. This included 33 uncharacterized (Table 1) and nine known TFs (*aflR*, *afoA*, *apdR*, *atnN*, *dbaA*, *mdpA*, *mdpE*, *pbcR* and *scpR*). Additionally, TF genes outside of the annotated secondary metabolite BGCs but previously associated with SM regulation based on publications (*rsmA*, *napA*, *mtfA*, and *hapX*) and information from the AspGD database (AN2025, AN2026, AN2241, AN4933 and AN8645) were also included. In total, 51 TFs (see Table 1 for details) were selected. These TFs are lowly- or non-expressed under most experimental conditions based on expression analysis of 878 public RNAseq data (Supplementary Fig. 1)[28], and their lack of expression is presumably the main reason for the silencing of SM clusters.

Building on the success of the AflR-OE and AN6790-OE strains described above, and in anticipation of scaling up for secondary metabolite purification, the same liquid culture setup was applied to the OE strains. When the expression of SM TFs was induced with xylose, the culture media of many strains exhibited diverse pigmentation distinct from the wild-type strain (Fig. 1e). The colors ranged from dark purple/red (e.g., RsmA and DbaA) to brown (e.g., AN2025, AN3269, AN8391, and AN5609) to light yellow (e.g., AflR, AN3501, AN8645, AN2241, AN3391, ApdR, AN10295, AN0533, AN4837, AtnN, and AN8506). Notably, some pigments were secreted into the broth, while others were contained within the mycelial mass (Fig. 1e). For example, the broth of fourteen OE strains (AN4933, NapA, AN7923, AN8902, AN7061, AN8509, AN9013, AN0153, AN1678, MtfA, AN3502, AfoA, AN7921, and AN10294) showed similar coloration to the WT, while their mycelia exhibited different colors. Lastly, a group of OE strains produced similar colors in both the mycelia and the culture media as the wild-type strain. Considering that many fungal metabolites are pigmented[29], these observations indicate the production of diverse metabolites by these strains and the success of the SM TF OE strategy.

## The collection of metabolites contains anti-bacterial, anti-fungal and anti-cancer activities

To determine the pharmaceutical potential of *A. nidulans*, crude extracts of secreted metabolites from the SM TF OE strains were assayed for various bioactivities. The use of crude extracts allows identification of bioactivities from intermediate metabolites and byproducts rather than just the final product of a given secondary metabolite BGC. For

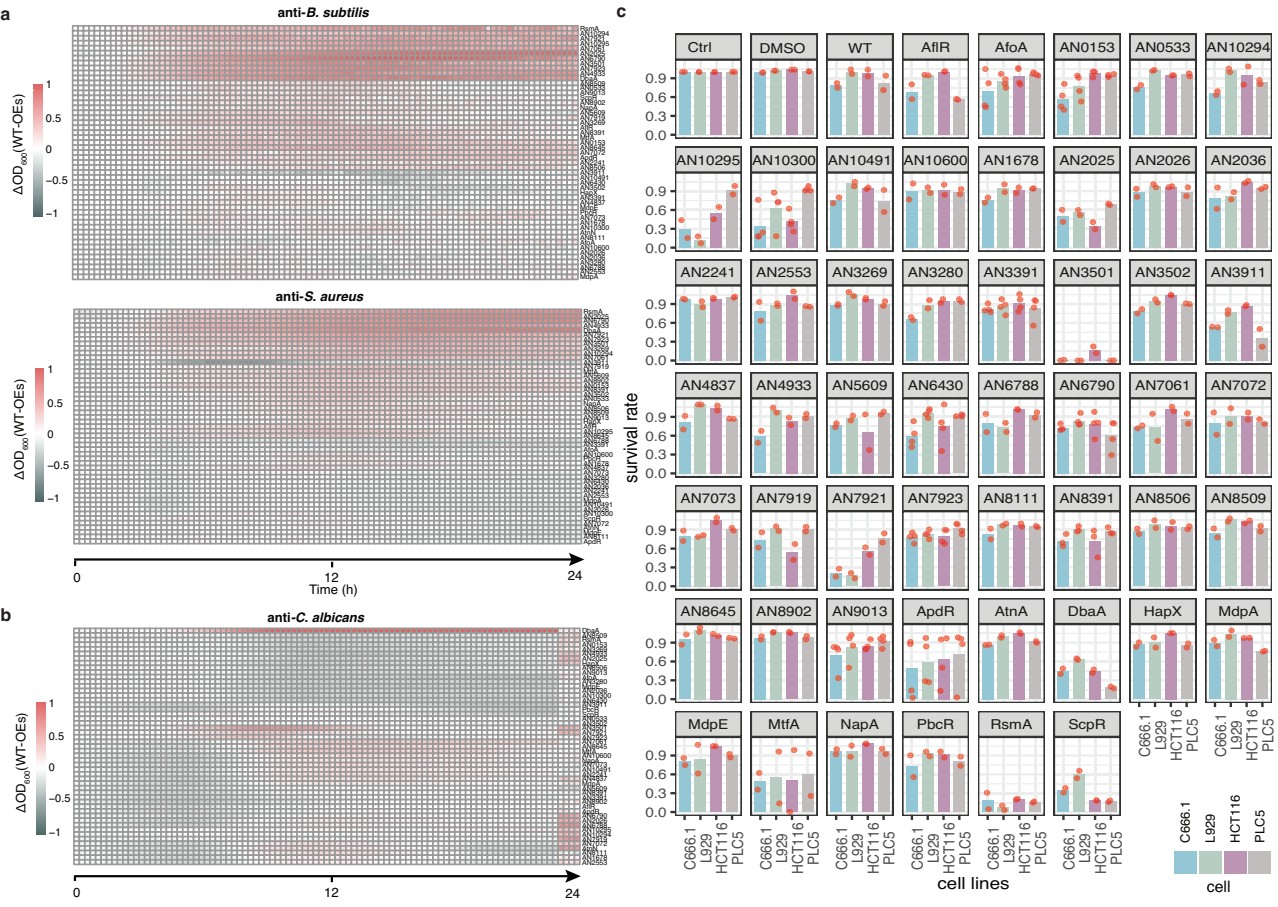

**Fig. 2 | Extracts of SM TF over-expression strains are rich in anti-bacterial, anti-fungal and anti-cancer activities.** A heatmap displaying the results of (**a**) *B. subtilis*, *S. aureus* (**b**) *C. albicans* growth assay in the presence and absence of EtOAc extracts from wild-type and the indicated over-expression strains. Each box displays the average $OD_{600}$ difference of *B. subtilis*, *S. aureus* or *C. albicans* incubated with the extract of OE strains relative to that of WT at indicated time points (Tables 2 and 3).

The presented result is an average of two biological repeats; **c** Bar charts showing the survival of murine fibroblast (L929), human colon cancer (HCT116), Alexander hepatoma carcinoma (PLC5), and Nasopharyngeal carcinoma (C666.1) cell lines. The mean of at least two biological repeats for each extract is shown, with the values of repeats depicted by red dots. Extract with less inhibition on the control cell line (L929) than on at least one cancer cell line was highlighted by a red box.

antibacterial activities, the extracts were tested against *Bacillus subtilis* and *Staphylococcus aureus*, representing non-pathogenic and pathogenic bacteria, respectively. Growth was measured by absorbance at $OD_{600}$ nm and compared between metabolites extracted from the culture media of wild-type and OE strains grown in the presence of 1% xylose. Remarkably, more than half of the SM TF OE strains produced metabolites with varying degrees of anti-bacterial activity against *B. subtilis* and/or *S. aureus* (Fig. 2a). One extract (AN3911) initially promoted bacterial growth between 4 and 10 h post-inoculation, but this effect diminished over time. Notably, the extracts of eight strains (DbaA, AN6790, AN3501, AN4933, AN7921, RsmA, AN7061, and AN2025) inhibited the growth of both *B. subtilis* and *S. aureus* by over 50% compared to the control, with the extract from the DbaA-OE strain exhibiting the most potent activity, achieving nearly 90% inhibition. This result aligns with previous findings that DbaA OE leads to the production of the antibiotic 2,4-dihydroxy-3-methyl-6-(2-oxopropyl)benzaldehyde (DHMBA)[30,31], validating our approach for identifying bioactive metabolites and highlighting the anti-bacterial potential of metabolites produced by the other seven OE strains.

Some fungal secondary metabolites possess anti-fungal properties that can be exploited for anti-fungal treatments, such as griseofulvin produced by *Penicillium griseofulvum*[31]. To this end, an anti-fungal bioactivity assay was performed using the same extracts on the opportunistic pathogen *Candida albicans* to identify anti-fungal properties. Only two extracts (from DbaA and AN7061 strains) produced a strong inhibitory effect ($\Delta OD_{600}$[WT-OE]

>0.5) on *C. albicans* growth (Fig. 2b), while the extract of the AN3911-OE strain, which had a positive effect on bacterial growth, also weakly promoted *C. albicans* growth. Interestingly, many extracts displayed mild ($\Delta OD_{600}$[WT-OE] <0.5) but significant anti-fungal activities with stage-specific patterns. For example, a group of extracts inhibited the early growth of *C. albicans* (~2–10 h post-inoculation), while other groups affected later stages (e.g., 6–24 h and 16–24 h). The biological significance of these stage-specific activities is currently unclear. Overall, there are less metabolites with anti-fungal activities comparing to anti-bacterial activities.

A high-throughput screen was also performed to identify anti-cancer activity. Cell lines representing three different cancer types (HCT116 colon cancer cells, PLC5 hepatoma cells, and C666.1 nasopharyngeal carcinoma cells) were tested, with the murine fibroblast cell line, L929, as the normal cell control. High content imaging analysis shows that the extracts exhibited differential inhibitory effects on mammalian cells (Fig. 2c). For HCT116 colon cancer cells, six extracts (from AN10300, DbaA, AN2025, AN3501, RsmA, and ScpR OE strains) demonstrated strong growth inhibition, resulting in a survival rate of less than 50% when treated with these extracts (Fig. 2c). For the hepatoma PLC5 cell line, treatment with extracts from five OE strains (AN3911, DbaA, AN3501, RsmA, and ScpR) led to a survival rate of less than 50% (Fig. 2c). For the C666.1 nasopharyngeal carcinoma cell line, eight extracts (from ApdR, AN10300, AN10295, DbaA, AN7921, AN3501, RsmA, and ScpR OE strains) displayed significant inhibition (Fig. 2c). Interestingly, the extract from the AN3911-OE strain, which positively affects bacterial and fungal growth, significantly inhibited the

proliferation of PLC5 and C666.1 cells but not HCT116 cells. Overall, six, five, and eight extracts showed potent anti-cancer activity against HCT116, PLC5, and C666.1 cancer cells, respectively, with three extracts (from AN3501, RsmA, and ScpR OE strains) inhibiting all three cancer cell lines. Among these extracts, a few (ones from the AN3501, RsmA, and AN10295 OE strains) were cytotoxic based on their effects on the normal fibroblast L929 cell line. The remaining six extracts (from AN10300, AN2025, AN3911, DbaA, ApdR, and ScpR OE strains) exhibited different levels of specificity against cancer cells (Fig. 2c, Supplementary Data 1), indicating their potential for anti-cancer drug development.

### Potential limitations, considerations and conclusions

Taken together, this study highlights the power of systematically activating BGCs in fungi to identify potential bioactive metabolites. Although our revised strategy—employing a stronger promoter and integration at the *yA* genomic locus—has proven successful for many BGCs, it does have some limitations. First, eukaryotic gene activation, and SM gene regulation in particular, is often subject to combinatorial control. This regulation requires multiple elements, including global TFs, chromatin-modifying proteins, mediators[32–34] and post-translational modifications. Our strategy of single TF OE does not account for this complex coordination and, therefore, could not activate all BGCs, which was elegantly demonstrated by the transcriptome analysis in the recent systematic SM TF OE study of *A. niger*[15]. A similar transcriptome analysis on our SM TF OE strains would help determine how many of the BGCs analyzed in this study could not be activated. Second, our approach primarily targets TFs located within annotated BGCs. As a result, metabolites produced by BGCs lacking a TF[35,36] or by non-clustered biosynthetic genes[10,37] may be overlooked. Third, excessive TF expression can lead to off-target activation, resulting in the production of non-specific metabolites and/or altered fungal physiology. While this may complicate the identification of primary BGC-derived metabolites, it could also be advantageous if these non-specific metabolites —typically not produced under normal conditions—exhibit useful bioactivities.

While the systematic TF OE strategy has demonstrated efficacy in *A. nidulans*, its broader application to non-model or genetically intractable fungi requires several important considerations. A key prerequisite is the availability of well-characterized genetic tools, including strong inducible promoters, efficient transformation systems, and reliable selection markers —resources often lacking in less-studied fungal species. Moreover, targeted genomic integration, which we achieved at the *yA* locus, may be inefficient or unpredictable in species with low homologous recombination efficiency or without an effective targeting system (e.g., *nkuA*Δ equivalent strains[38]). At the transcriptional level, BGCs in some fungi may reside in heterochromatic regions, where silencing mechanisms differ from those in *A. nidulans*[39], making TF overexpression alone insufficient to overcome repression. Additionally, promoter activity can be context-dependent. Even strong promoters like *xylP* may not function optimally under all conditions[40] or in other fungal species, presumably due to differences in transcriptional machinery or regulatory networks.

Despite these challenges, advances in synthetic biology and gene editing technologies—such as CRISPR-Cas—have made genome engineering increasingly feasible in fungi. Therefore, the SM TF OE strategy presented in this work offers a powerful approach for uncovering the pharmaceutical potential of diverse fungal species.

## Methods
### Strains construction

The xylose-inducible promoter *xylP*(p) from *P. chrysogenum*[23] was selected for conditional expression of SM TFs. To allow detection of SMTFs, epitope tags such as 3×HA, 3×FLAG, or 6×HIS were also added to the C-terminus of each SMTF of interest[41]. To create the OE constructs, a starting plasmid (XPyA-3, CWB13) carrying the *xylP* promoter, a 724 bp internal fragment of the *yA* gene for targeted integration[42], and the *Aspergillus fumigatus pyroA* gene as the selection marker[43], was linearized by SmaI (NEB, cat no. R0141)

digestion. Fragments containing epitope tag, 3×HA or 3×FLAG, or 6×HIS and *ADH1*(t) were then amplified from the established tagging plasmids using the TF gene-specific primer OE-F and ADH1(t)_R (1716) (CTA-GAACTAGTGGATCCCCCCCGGTAGAGGTGTGGTC), and after PCR purification, fused with the prepared linear XPyA-3 vector by isothermal assembly. In this way, a set of new starting plasmids containing different epitope tags was created, from which a linear vector (XPyA-3-1) carrying *xylP* promoter, 3×HA and the terminator *ADH1*(t) was amplified by inverse PCR using oligo pks_reverse_R (2661) and longtin F (3789). To create target gene OE plasmid, a fragment starting from the 10 bp upstream of start codon ATG to the base exactly before the stop codon of target gene was amplified from genomic DNA of the *A. nidulans* MH11036 strain and fused with XPyA-3-1, which efficiently resulted in production of an OE plasmids library composed of genes-of-interest fusing with 3×HA epitope tags (Supplementary Fig. 2a). Likewise, OE plasmids for target genes with other epitope tags, 3×FLAG and 6×HIS, were also created. The plasmids were confirmed by sequencing and transformed into the MH11036 (*veA1*, *nkuA*Δ, *pyroA4*, *riboB2*) strain. Transformation was carried out using the protoplast method[44]. The OE constructs carry a 724 bp internal fragment of the *yA* gene and the *A. fumigatus pyroA* gene for targeted integration[24] and selection, respectively. If the OE plasmid integrated at the *yA* locus, the transformants would have the *yA* gene disrupted, leading to yellow conidia (Supplementary Fig. 2b). Therefore, transformants with yellow conidia were selected on solid minimal medium lacking pyridoxine. Southern blot analysis was applied to confirm integration at *yA*, determine the construct copy-number integrated into the genome, and rule out any ectopic integration that might have happened other than at the *yA* locus (Supplementary Fig. 3). Transformants with only a single integrated plasmid copy were selected for further study.

### Growth conditions

*Aspergillus nidulans* minimal (ANM) media was used for standard growth[45]. For SM TF OE, $5 \times 10^5$ mL$^{-1}$ spores of SM TF OE strains and their wild-type recipient strain (*pyroA4*, *riboB2*, and *nkuA*Δ) were grown in 100 mL liquid ANM medium supplemented with pyridoxine (0.5 mg L$^{-1}$) (Sigma, cat. no. P9755) and riboflavin (2.5 mg L$^{-1}$) (Sigma, cat. no. R7649) at 37 °C with shaking at 220 rpm. After 48 h, xylose was added with a final concentration being 1% and strains were grown for an additional three days.

### Western blot analysis

Total proteins were extracted using the TCA method[46] and quantified using BioRAD DC Protein Assay (BIO-RAD, cat. no. 5000112). 50 µg total protein was loaded in SDS-PAGE gel for each sample. The transformation of the protein in gel to PVDF membrane (BIO-RAD, cat. no. 1620177) were performed by wet-transfer overnight at 40 V. With respect to primary antibody, the dilution is 1:3000 for HA-Tagged protein (Santa Cruz, cat. no. sc7392), 1:5000 for FLAG-tagged protein (Monoclonal ANTI-FLAG® M2, Sigma, cat. no. F3165), 1:5000 for HIS-tagged protein (Abcam, ab9108) and 1:10000 for H3 (Abcam, ab1791). 1:5000 horseradish peroxidase-conjugated anti-mouse (Sigma-Aldrich, AP124P) or anti-rabbit antibodies (Sigma-Aldrich, AP132P) were used based on the primary antibody type. The Clarity™ ECL substrate kit (BIO-RAD, cat. no. 1705060) was applied for chemiluminescence detection. The uncropped Western Blot images of Fig. 1a are presented in Supplementary Fig. 4.

### Metabolites extraction

Culture media was collected by filtering using a clean cheesecloth to remove mycelia and extracted twice with equal volume of EtOAc. The ethyl acetate (EtOAc) extracts were dried by rotary evaporator (EYELA, Japan) at 25 °C. The residues were resuspended in dimethyl sulfoxide (DMSO) (ChemCruz, cat. no. SC-358801) to bring the concentration to 10 mg mL$^{-1}$.

### LC-MS analysis

LC-MS was performed using the ion trap mass spectrometer in positive ion mode with a BEH C18 Column, 130 Å, 1.7 µm, 2.1 × 100 mm (Waters,

**Table 2 | Summary of bacterial inhibition tests**

| Extract | B. subtilis | | | S. aureus | | |
|---|---|---|---|---|---|---|
| | Average OD$_{600}$ at 16 h | Survival rate (%) | p-val[a] | Average OD$_{600}$ at 16 h | Survival rate (%) | p-val[a] |
| WT | 1.4 | 100.5 | 0.744 | 1.3 | 101.7 | 0.087 |
| DMSO | 1.3 | 100.9 | 0.787 | 1.2 | 91.9 | 0.003 |
| H$_2$O | 1.3 | 99.3 | 0.85 | 1.3 | 98.5 | 0.188 |
| DbaA | 0.2 | 11.9 | 0 | 0.2 | 13.8 | 0 |
| AN6790 | 0.2 | 17.8 | 0 | 0.2 | 17 | 0 |
| AN3501 | 0.3 | 19.7 | 0 | 0.5 | 39.3 | 0 |
| AN4933 | 0.3 | 21 | 0 | 0.3 | 23.2 | 0 |
| AN7921 | 0.3 | 22.6 | 0 | 0.4 | 30.5 | 0 |
| RsmA | 0.4 | 26.8 | 0 | 0.4 | 28.7 | 0 |
| AN7061 | 0.4 | 27 | 0 | 0.5 | 34.3 | 0 |
| AN2025 | 0.5 | 36.1 | 0 | 0.6 | 43.2 | 0 |
| AN10295 | 0.6 | 43.4 | 0 | 0.7 | 52.1 | 0 |
| AN0153 | 0.7 | 50.6 | 0 | 0.7 | 51.9 | 0 |
| MtfA | 0.7 | 52.5 | 0.001 | 0.7 | 55.7 | 0 |
| AN7923 | 0.7 | 53.9 | 0 | 0.8 | 59.7 | 0 |
| AN10294 | 0.8 | 58.2 | 0 | 0.6 | 44.7 | 0 |
| AN8391 | 0.8 | 62 | 0.001 | 0.6 | 47.2 | 0 |
| AN7919 | 0.9 | 64.6 | 0.001 | 0.8 | 64.1 | 0.007 |
| AN8645 | 0.9 | 69.7 | 0.001 | 0.9 | 72.3 | 0.001 |
| AN3269 | 1 | 70.7 | 0 | 0.8 | 62.4 | 0 |
| AN8506 | 1 | 76.6 | 0 | 0.7 | 55.7 | 0 |
| AN8902 | 1 | 77.3 | 0.002 | 0.7 | 53.9 | 0 |
| AN9013 | 1.1 | 78.4 | 0 | 0.9 | 74.3 | 0.005 |
| AN5609 | 1 | 80.2 | 0.008 | 1.1 | 85.6 | 0.002 |
| AflR | 1.1 | 81.4 | 0.004 | 1 | 80 | 0.01 |
| AN0533 | 1.1 | 81.6 | 0 | 1 | 77.7 | 0 |
| AN8509 | 1.1 | 81.6 | 0.001 | 0.9 | 70.9 | 0.002 |
| AN4837 | 1.1 | 86.5 | 0.017 | 1 | 74.1 | 0.005 |
| ScpR | 1.2 | 91.6 | 0.001 | 1.2 | 96 | 0.047 |
| AN1678 | 1.2 | 93.1 | 0.113 | 0.9 | 65.9 | 0.005 |
| AN7073 | 1.2 | 93.2 | 0.097 | 1.2 | 94.5 | 0.002 |
| HapX | 1.3 | 93.7 | 0.004 | 1.2 | 93.6 | 0.002 |
| MdpE | 1.3 | 94.3 | 0.003 | 1.3 | 99.2 | 0.444 |
| AN2036 | 1.3 | 94.5 | 0.004 | 1.2 | 95.1 | 0.037 |
| AtnN | 1.2 | 94.8 | 0.174 | 1.1 | 83.3 | 0 |
| PbcR | 1.2 | 96.3 | 0.317 | 1.3 | 99.4 | 0.752 |
| AN7072 | 1.2 | 96.3 | 0.474 | 1.1 | 87.2 | 0.001 |
| AN8111 | 1.3 | 96.8 | 0.379 | 1.3 | 96.6 | 7.04 |
| AN10300 | 1.3 | 96.8 | 0.021 | 1.2 | 92.2 | 0.258 |
| AN3280 | 1.3 | 97.4 | 0.049 | 1.2 | 97.2 | 0.029 |
| AN6430 | 1.3 | 98.4 | 0.32 | 1.3 | 98.6 | 0.494 |
| AN3502 | 1.3 | 98.8 | 0.898 | 0.8 | 61.6 | 0 |
| AN2553 | 1.3 | 98.9 | 0.742 | 1.3 | 95.3 | 0.002 |
| MdpA | 1.3 | 99.3 | 0.844 | 1.3 | 96.5 | 0.076 |
| ApdR | 1.3 | 99.5 | 0.9 | 1.3 | 96.9 | 0.021 |
| NapA | 1.3 | 99.9 | 0.996 | 1.1 | 82.5 | 0.002 |
| AfoA | 1.4 | 100 | 0.956 | 1.3 | 101.3 | 0.237 |
| AN2026 | 1.3 | 100.4 | 0.9 | 1.3 | 102 | 0.095 |

**Table 2 (continued) | Summary of bacterial inhibition tests**

| Extract | B. subtilis | | | S. aureus | | |
|---|---|---|---|---|---|---|
| | Average OD$_{600}$ at 16 h | Survival rate (%) | p-val[a] | Average OD$_{600}$ at 16 h | Survival rate (%) | p-val[a] |
| AN10600 | 1.3 | 101.7 | 0.711 | 1.2 | 89.5 | 0.002 |
| AN10491 | 1.3 | 101.8 | 0.681 | 1.4 | 103.8 | 0.131 |
| AN2241 | 1.3 | 102 | 0.567 | 1.3 | 97 | 0.033 |
| AN6788 | 1.3 | 103.7 | 0.309 | 1.4 | 103.3 | 0.019 |
| AN3391 | 1.3 | 103.7 | 0.317 | 1.3 | 101.3 | 0.269 |
| AN3911 | 1.4 | 106.2 | 0.003 | 1.3 | 104.2 | 0.066 |

[a]Two tail t-test.

USA), as described in the previous study[40]. Each sample with a concentration of 10 mg mL$^{-1}$ was diluted tenfold with methanol and loaded in triplicates. Waters Masslynx V4.1 workstation software and the program MestReNova (Version 14.1.0, MestReC, Spain) was used to process the LC-MS data.

**Anti-bacterial assay**

To detect the antibacterial activity of EtOAc extracts from each OE strain, the gram-positive bacteria *S. aureus* (ATCC 25904) and *B. subtilis* (ATCC 21336) were used. Bacterial cells were grown in liquid LB medium in a microtitre plate. Extracts of OE strains were added at a final concentration of 50 μg mL$^{-1}$. The optical density of bacteria at $\lambda = 600$ nm (OD$_{600}$) was measured at 15 min intervals over a 24 h period using a Cytation 3 microplate reader. Each extract was tested against each bacterial species three times and repeated twice independently (i.e. a total of six measurements). The differences in the OD$_{600}$ values of the test bacteria incubated with extracts from WT and each OE strain were presented as a heatmap (Fig. 2b).

**Anti-fungal assay**

To test the antifungal activity of these EtOAc extracts, *C. albicans* (SC5314) was used as the test strain. It is an opportunistic pathogenic yeast that poses a threat to immunocompromised individuals. The screening test was performed in a microplate assay and 5 μL diluted EtOAc extracts were added to 195 μL diluted *C. albicans* (OD$_{600} = 0.01$) at logarithmic phase to give a final concentration of 50 μg mL$^{-1}$. The OD$_{600}$ of *C. albicans* during the 24 h growth period was measured at 15 min intervals. The test for each EtOAc extract against each strain was performed in triplicate and repeated twice independently (i.e. six measurements in total). The difference in the OD$_{600}$ of *C. albicans* culture incubated with EtOAc extracts from WT and each OE strain was visualized as a heatmap (Fig. 2c).

**Anti-cancer cell assay**

Cell viability screening was conducted on the EtOAc extracts using L929, HCT116, PLC5, and C666.1 cell lines, which were routinely subjected to mycoplasma testing using the mycoplasma detection kit from TransGen Biotech (FM311-01). Cells were seeded in 96-well plates at the following densities: L929 ($1.5 \times 10^4$), HCT116 ($2.5 \times 10^4$), PLC5 ($1.0 \times 10^4$), and C666.1 ($4.0 \times 10^4$). L929, HCT116, and PLC5 cells were cultured in Dulbecco's modified Eagle's medium (Gibco, cat. no. 12100046) with 10% fetal bovine serum (FBS) (Gibco, cat. no. 10270106), while C666.1 cells were cultured in RPMI 1640 medium with GlutaMAX (Gibco, cat. no. 61870036) and 10% FBS. For the treatment of L929 and HCT116 cells, extracts were applied at a final concentration of 3 μg mL$^{-1}$ for 24 h. For PLC5 and C666.1 cells, extracts were used at a final concentration of 1.5 μg mL$^{-1}$ for 48 h. All cells were maintained in a humidified incubator at 37 °C with 5% CO2. After treatment, the cells were stained with the LIVE/DEAD$^{TM}$ Viability/Cytotoxicity Kit (Invitrogen, cat. no. L3224) and Hoechst 33342 (Invitrogen, cat. no. H3570), then imaged using the PerkinElmer Opera

## Table 3 | Summary of *C. albicans* growth test results

| Extract | C. albicans growth assay | | |
|---|---|---|---|
| | Average (OD$_{600}$ at 16 h) | Survival rate (%) | p-val[a] |
| WT | 1.9 | 106.7 | 0.004 |
| H$_2$O | 1.7 | 98.8 | 0.139 |
| DbaA | 0.2 | 14.1 | 0 |
| AN7061 | 1.1 | 79 | 0.001 |
| MtfA | 1.3 | 93.1 | 0.191 |
| AN3501 | 1.7 | 94.8 | 0.002 |
| AN7921 | 1.7 | 95.1 | 0 |
| AN8902 | 1.3 | 95.6 | 0.191 |
| AN8391 | 1.3 | 96.3 | 0.178 |
| RsmA | 1.7 | 97.2 | 0.05 |
| AN8509 | 1.7 | 98.3 | 0.187 |
| AN8506 | 1.7 | 98.4 | 0.083 |
| AN5609 | 1.4 | 98.5 | 0.594 |
| AN8645 | 1.4 | 99.1 | 0.719 |
| AN9013 | 1.7 | 99.1 | 0.487 |
| AN10600 | 1.4 | 100 | 0.988 |
| AN3269 | 1.8 | 100.2 | 0.698 |
| AN0533 | 1.8 | 100.2 | 0.821 |
| AflR | 1.4 | 100.7 | 0.836 |
| HapX | 1.8 | 100.7 | 0.349 |
| ApdR | 1.4 | 101.4 | 0.581 |
| AN3502 | 1.8 | 101.5 | 0.018 |
| AN2241 | 1.4 | 101.7 | 0.571 |
| AN2036 | 1.8 | 101.8 | 0.024 |
| AN2025 | 1.8 | 101.9 | 0.034 |
| AN0153 | 1.8 | 103.4 | 0.033 |
| AN3391 | 1.4 | 103.5 | 0.194 |
| AN7923 | 1.8 | 103.7 | 0.001 |
| AtnN | 1.4 | 103.8 | 0.664 |
| NapA | 1.4 | 104.2 | 0.359 |
| AN10491 | 1.5 | 104.4 | 0.121 |
| AN7073 | 1.5 | 104.7 | 0.101 |
| AN10294 | 1.8 | 105.1 | 0.026 |
| ScpR | 1.8 | 105.2 | 0.364 |
| AN4837 | 1.5 | 105.2 | 0.135 |
| AN7072 | 1.5 | 105.4 | 0.073 |
| MdpE | 1.9 | 105.6 | 0.001 |
| AN6430 | 1.9 | 105.7 | 0.102 |
| AN2553 | 1.5 | 105.8 | 0.207 |
| AN1678 | 1.5 | 106.3 | 0.117 |
| AN7919 | 1.9 | 106.4 | 0.001 |
| AN8111 | 1.5 | 107 | 0.072 |
| AN3280 | 1.9 | 107 | 0.004 |
| MdpA | 1.5 | 107.1 | 0.037 |
| AN4933 | 1.9 | 107.2 | 0 |
| AN6788 | 1.5 | 107.2 | 0.035 |
| AfoA | 1.9 | 107.6 | 0.001 |
| AN10295 | 1.5 | 107.8 | 0.021 |
| AN6790 | 1.5 | 108.1 | 0.083 |
| AN10300 | 1.9 | 108.2 | 0.094 |

## Table 3 (continued) | Summary of *C. albicans* growth test results

| Extract | C. albicans growth assay | | |
|---|---|---|---|
| | Average (OD$_{600}$ at 16 h) | Survival rate (%) | p-val[a] |
| AN2026 | 1.5 | 110.7 | 0.092 |
| PbcR | 1.7 | 120 | 0.023 |
| AN3911 | 1.8 | 130.9 | 0 |

[a]Two tail t-test.

Phenix High Content Screening System in confocal mode with a 20×water objective (NA 1.0). The fluorophores were detected using the following excitation and emission (Ex/Em) wavelengths: Hoechst 33342 (405/435–480), calcein-AM (488/500–550), and ethidium homodimer-1 (561/570–630). Image analysis was conducted using PerkinElmer Harmony Software.

### Statistics and reproducibility
For the same TF OE strain, two biological replicates were performed on two different days for EtoAc extract preparation. Anti-bacterial assay against *B. subtilis* and *S. aureus*, and anti-fungal assay against *C. albicans* were performed with two biological replicates, each with three technical replicates. In the anti-cancer cell assay, a minimum of two biological replicates were analyzed, with nine selected strains subjected to four replicates (see Supplementary Data 1). For statistical analyses presented in Tables 2 and 3, two-tailed student *t*-test was employed to compare each OE strain with the WT cultured in ANM medium. Differences were considered statistically significant at $p < 0.05$, very significant at $p < 0.01$, and extremely significant at $p < 0.001$.

### Reporting summary
Further information on research design is available in the Nature Portfolio Reporting Summary linked to this article.

### Data availability
All data generated or analyzed in this study have been included in the supplementary files. Source data can be found in Supplementary Data 1 and the original Western blots are shown in Supplementary Fig. 4. The plasmids and strains will be made available on request. Please contact the corresponding authors at koonhowong@um.edu.mo and guosh@wxu.edu.cn.

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

## Acknowledgements

We thank the members of the Wong laboratory for their valuable comments and discussions, and Prof. Yit-Heng Chooi and Prof. Zhou (Alex) Shang for their advice and suggestions. We acknowledge the services and technical support provided by the Genomics and Single Cell Analysis Core, as well as the Drug and Development Core of the Faculty of Health Sciences at the University of Macau. This work was partially conducted on the High-Performance Computing Cluster (HPC), supported by the Information and Communication Technology Office (ICTO) of the University of Macau. We gratefully acknowledge funding support from the Science and Technology Development Fund, Macao S.A.R. (FDCT, project no. 0099/2022/A2), the Research Services and Knowledge Transfer Office (project nos. MYRG2022-00107-FHS and MYRG-GRG2023-00084-FHS-UMDF), and the Faculty of Health Sciences, University of Macau, awarded to K.H.W. and Research Grants Council, HK. (C5012-15E) to B.C.B.K.

## Author contributions

S.G. and K.H.W. conceived the study, designed experiments, and interpreted data. S.G., L.Q., C.Y.C., L.F. and C.C.M. performed experiments. S.G., L.P., C.P. and X.L. performed Bioinformatics analysis. K.T. and Z.D. provided technical support. K.H.W., K.T. and B.C.B.K. provided funding. S.G. and K.H.W. wrote the manuscript.

## Competing interests

The authors declare no conflict of interest. K.H.W. is an Editorial Board Member for *Communications Biology*, but was not involved in the editorial review of, nor the decision to publish this article.
