## [Transparent Peer Review file · Communications Biology]

Systematic over-expression of secondary metabolism transcription factors to reveal the pharmaceutical potential of *Aspergillus nidulans*

Corresponding Author: Professor Koon Ho Wong

Version 0:

Reviewer comments:

Reviewer #1

(Remarks to the Author)

The starting hypothesis of the manuscript is that the level of expression (referred as sufficient overexpression) of transcription factors is key in the activation of biosynthetic gene clusters. The authors used a tunable strong promoter and a targeted location (yA gene locus) in the *A. nidulans* genome to overexpress 51 TFs (total). Although some BGCs were expressed and secondary metabolites were produced, there is no experimental evidence (RNA-seq or qPCR or any other method for transcript analysis) in the manuscript showing the impact of the expression levels of the TFs on the BGCs activation.

Ultimately, the major claim of the manuscript is the systematic activation of BGCs and the production of SMs by TFs overexpression which is interesting and has already been done in other fungi (<https://doi.org/10.1101/2024.07.18.604165>) with the experimental support of transcriptomic..

Major comments

Line 81: How did the authors decided which TF is located within a BGC?

Line 98-101: Liquid medium is not the best way to evaluate pigmentation. a growth profile on agar plate would be a better method both for secreted SM/pigments and for SM or pigments retained in the mycelium. Please justify your choice for this method.

Line 108-109: what is the rational for choosing these two bacteria? gram - and gram +? Please develop.

Line 126 and 128: please define "strong" and "mild"

Line 131-134: "While the biological...fungi" this sentence is not clear and I do not see evidence or support for it.

Methods section: The fungal transformation is not described. Please add details on the procedure.

Minor comments

Line 174: correction, start codon and stop codon instead of "code"

Line 177: I suggest to revise the English in this sentence, are the authors refering to the "genes of interest"? Please clarify.

Reviewer #2

(Remarks to the Author)

The manuscript by Guo et al. describes a strategy involving the overexpression of transcription factor genes embedded in biosynthetic gene clusters to reveal the secondary metabolites produced by fungi. The manuscript is brief and to the point and the work is of interest and importance.

Comments:

Overall, the manuscript is very brief. While I appreciate the authors' successful efforts to be succinct, at times the manuscript is too succinct. There are three areas where clarity could be improved:

a) Abstract: too short and generic. It needs to be expanded so that readers can understand what the work is about. For example, the authors talk of a "systematic approach" but it is not clear to the reader what that approach is.

b) The main text is one long section. I think it would be easier for readers if the authors used headings and titles of the different sections of their main results.

c) I missed a paragraph of the caveats / shortcomings of the authors' approach. While undoubtedly powerful, the approach is mostly suited to fungal organisms that are genetically tractable. What would be the considerations one would have to take into account if they were interested in implementing this approach for a non-model or less well-studied fungus?

Lines 33-34: it is important to emphasize here that the vast majority of the BGCs identified in fungal genomes are "putative" and lack functional validation. In other words, it is not only that "most remain transcriptionally silent" but also that we lack clarity on how many of them are actually functional.

Line 38: a few citations of these low-throughput methods would be helpful here.

Line 39: given that the authors' approach is only applicable to clusters with an embedded transcription factor gene, pitching this approach as a way "to evaluate the overall pharmaceutical potential" is somewhat of an exaggeration.

Lines 84-90: the numbers don't add up here: 38 uncharacterized TFs + 11 known TFs + 4 TFs outside of BGCs based on publications + 7 TFs outside of BGCs based on AspGD information = 60. However, the authors say that "In total, 51 TFs were selected". Additionally, it would be useful to include this information on Table 1.

Line 106: "assay" -> "assayed"

Lines 131-134: The authors have only tested against 2 bacteria and 1 fungus. There is also no information on whether *A. nidulans* does (or does not) naturally interact with these three species. I just don't see how these data remotely justify the hypothesis that there is "a selective pressure for *A. nidulans* to evolve metabolites for competition against bacteria over other fungi". I suggest deletion.

Line 221: please state the bacterial strains tested.

Line 230: please state the fungal strain tested.

Version 1:

Reviewer comments:

Reviewer #2

(Remarks to the Author)

The authors have done a good job addressing my comments and suggestions. I have no further requests.

Point-by-Point Response to Editor's and Reviewers' Comments
COMMSBIO-25-4480-T

Reviewer #1 (Remarks to the Author):

The starting hypothesis of the manuscript is that the level of expression (referred as sufficient overexpression) of transcription factors is key in the activation of biosynthetic gene clusters. The authors used a tunable strong promoter and a targeted location (yA gene locus) in the *A. nidulans* genome to overexpress 51 TFs (total). Although some BGCs were expressed and secondary metabolites were produced, there is no experimental evidence (RNA-seq or qPCR or any other method for transcript analysis) in the manuscript showing the impact of the expression levels of the TFs on the BGCs activation. Ultimately, the major claim of the manuscript is the systematic activation of BGCs and the production of SMs by TFs overexpression which is interesting and has already been done in other fungi (<https://doi.org/10.1101/2024.07.18.604165>) with the experimental support of transcriptomic.

Response:

Thank you for taking the time to review our work and for your helpful comments, which we have carefully addressed as detailed below.

For the comment about experimental evidence showing the impact of the expression levels of the TFs on the BGCs activation, we have in fact performed transcription profiling analysis but feel that the description of those data is a different focus from what we hope to highlight in this paper. Therefore, we prefer to present them in a separate manuscript, which is under preparation.

We appreciate your positive feedback and thank you for pointing out the reference, which we have now described and cited in various sections of our manuscript:

- Lines 45-48: “Notably, a recent study has systematically over-expressed 58 cluster-specific transcription factors (TFs) of *Aspergillus niger*, leading to the production of potentially novel compounds¹⁵. Although the bioactivity of those compounds was not determined in that study, such systematic approach provides a valuable means to evaluate the pharmaceutical potential of a given fungus.”
- Lines 61-62: “Upon examining the experimental setup of the *A. nidulans* study¹⁷ and comparing with the *A. niger* study¹⁵, a potential explanation for the low success rate emerged.”
- Lines 80-82: “To determine if the over-expression could successfully induce SM production and to compare with the previous studies^{15, 25}, we used liquid culture conditions to analyze and compare metabolites production.”
- Lines 180-186: “First, eukaryotic gene activation, and SM gene regulation in particular, is often subject to combinatorial control. This regulation requires multiple elements, including global TFs, chromatin-modifying proteins, mediators³³⁻³⁵ and post-translational modifications. Our strategy of single TF OE does not account for this complex coordination and, therefore, could not activate all BGCs, which was nicely demonstrated by the transcriptome analysis in the recent systematic SM TF OE study of *A. niger*¹⁵. A similar transcriptome analysis on our SM TF OE strains would help determine how many of the BGCs analyzed in this study could not be activated.”

Major comments

Line 81: How did the authors decided which TF is located within a BGC?

Response:

Two ways were used to identified TFs located within a BGC. First, we searched the annotated secondary metabolite BGCs predicted by the SMURF (Secondary Metabolite Unknown Regions Finder) program (Khaldi et al., 2010) for transcription factor genes. This yields 33 transcription factor genes. Second, we included nine transcription factor genes (*aflR*, *afoA*, *apdR*, *atnN*, *dbaA*, *mdpA*, *mdpE*, *pbcR* and *scpR*) that reside within experimentally proven BGCs. The details of the transcription factors can be found in Table 1.

Khaldi, N. et al. SMURF: Genomic mapping of fungal secondary metabolite clusters. *Fungal Genet Biol* **47**, 736-741 (2010).

Line 98-101: Liquid medium is not the best way to evaluate pigmentation. a growth profile on agar plate would be a better method both for secreted SM/pigments and for SM or pigments retained in the mycelium. Please justify your choice for this method.

Response:

We initially designed the experiment setup to make a direct comparison to the study of Ahuja et al., 2012, in which liquid medium was used. In term of SM production, liquid and solid media have their own advantages and have been used by different studies / purposes. Our main purpose is to obtain crude metabolites for bioactivity screening. We think that either method (liquid or solid) should be fine and liquid culture is easier to handle and process as well as to scale up for SM purification (if necessary) than solid plate culture. Therefore, we chose liquid culture as the method of choice.

We agree with this reviewer that this should be clarified and have added the following justifications in the revised manuscript:

- Lines 80-82: “*To determine if the over-expression could successfully induce SM production and compare with the previous studies^{15, 25}, we used liquid culture conditions to analyze and compare metabolites production*”
- lines 109-111 “*Building on the success of the AflR-OE and AN6790-OE strains described above, and in anticipation of scaling up for secondary metabolite purification, the same liquid culture setup was applied to the OE strains.*”.

15. Semper, C. et al. Global survey of secondary metabolism in *Aspergillus niger* via activation of specific transcription factors. bioRxiv, 2024.2007. 2018.604165 (2024).

25. Ahuja, M. et al. Illuminating the diversity of aromatic polyketide synthases in *Aspergillus nidulans*. *J Am Chem Soc* **134**, 8212-8221 (2012).

Line 108-109: what is the rationale for choosing these two bacteria? gram - and gram +? Please develop.

Response:

Bacillus subtilis is a non-pathogenic model bacterium that is easy to work with and ideal for bioactivity screening, while *Staphylococcus aureus* is a leading human pathogen known for causing serious infections and developing antibiotic resistance. Therefore, we consider them ideal candidates for antibacterial activity screening.

The related information has been included in the lines 128-130: “*For antibacterial activities, the extracts were tested against Bacillus subtilis and Staphylococcus aureus, representing non-pathogenic and pathogenic bacteria, respectively.*”.

Line 126 and 128: please define “strong” and “mild”

Response:

The definition of strong and mild has been added – ($\Delta OD_{600}[\text{WT-OE}] > 0.5$) and ($\Delta OD_{600}[\text{WT-OE}] < 0.5$), respectively. See lines 147 and 150 in the revised version.

Line 131-134: “While the biological...fungi” this sentence is not clear and I do not see evidence or support for it.

Response:

We have rephrased the sentence and removed the statement “selective pressure for *A. nidulans* to evolve metabolites for competition against bacteria over other fungi”. The revised version (lines 152-154) now reads as “*The biological significance of these stage-specific activities is currently unclear. Overall, there are less metabolites with anti-fungal activities comparing to anti-bacteria activities.*”

Methods section: The fungal transformation is not described. Please add details on the procedure.

Response:

The information has been added. Please see lines 230-237: “*The plasmids were confirmed by sequencing and transformed into the MH11036 (veA1, nkuAΔ, pyroA4, riboB2) strain. Transformation was carried out as described previously⁴⁵. The OE constructs carry a 724 bp internal fragment of the yA gene and the Aspergillus fumigatus pyroA gene for targeted integration²⁴ and selection, respectively. If the OE plasmid integrated at the yA locus, the transformants would have the yA gene disrupted, leading to yellow conidia (Supplementary Figure 2b). Therefore, transformants with yellow conidia were selected on solid minimal medium lacking pyridoxine.*”

24. Wong, K.H., Hynes, M.J., Todd, R.B. & Davis, M.A. Transcriptional control of *nmrA* by the bZIP transcription factor MeaB reveals a new level of nitrogen regulation in *Aspergillus nidulans*. *Mol Microbiol* 66, 534-551 (2007).

45. Andrianopoulos, A. & Hynes, M.J. Cloning and analysis of the positively acting regulatory gene *amdR* from *Aspergillus nidulans*. *Mol Cell Biol* 8, 3532-3541 (1988).

Minor comments

Line 174: correction, start codon and stop codon instead of “ code”

Response:

Corrected.

Line 177: I suggest to revise the English in this sentence, are the authors referring to the “ genes of interest”? Please clarify.

Response:

Corrected to “genes-of-interest”.

Reviewer #2 (Remarks to the Author):

The manuscript by Guo et al. describes a strategy involving the overexpression of transcription factor genes embedded in biosynthetic gene clusters to reveal the secondary metabolites produced by fungi. The manuscript is brief and to the point and the work is of interest and importance.

Response:

Thank you for your time and helpful comments. We have addressed all your comments as detailed below.

Comments:

Overall, the manuscript is very brief. While I appreciate the authors' successful efforts to be succinct, at times the manuscript is too succinct. There are three areas where clarity could be improved:

a) Abstract: too short and generic. It needs to be expanded so that readers can understand what the work is about. For example, the authors talk of a "systematic approach" but it is not clear to the reader what that approach is.

Response:

The abstract has been expanded (underlined below) to include details about the systematic approach, as follows: *“Many life-saving drugs are derived from fungal secondary metabolites, and the rich diversity of these metabolites is a gold mine of bioactive compounds for drug discovery. However, the biosynthetic genes for most secondary metabolites remain transcriptionally silent in fungi, posing a significant bottleneck in their discovery. Here, we apply a systematic approach to separately over-express 51 secondary metabolite (SM)-related transcription factors (TFs) using a strong xylose-inducible promoter of the *Penicillium chrysogenum* xylP gene. Growing the individual SM TF over-expression strains in the presence of xylose leads to the production of a collection of diverse metabolites with anti-bacterial, anti-fungal and anti-cancer activities. The overall approach and the over-expression system established in this study are broadly-applicable, providing a valuable means to revealing the pharmaceutical potentials of fungi.”*

b) The main text is one long section. I think it would be easier for readers if the authors used headings and titles of the different sections of their main results.

Response:

We have added the following headings and titles of the different sections:

Line 60 – ***“Expression of SM TF from a strong promoter facilitates SM cluster gene activation.”***

Lines 93-94 – ***“Systematic overexpression of transcription factors implicated in secondary metabolism leads to production of diverse metabolites.”***

Line 124 – ***“The collection of metabolites contains anti-bacterial, anti-fungal and anti-cancer activities.”***

Line 176 – ***“Potential limitations, considerations and conclusions.”***

c) I missed a paragraph of the caveats / shortcomings of the authors' approach. While undoubtedly powerful, the approach is mostly suited to fungal organisms that are genetically tractable. What would be the considerations one would have to take into account if they were interested in implementing this approach for a non-model or less well-studied fungus?

Response:

We have added a section about the shortcomings of our approach and considerations for a non-model or less well-studied fungi. The added section is in lines 176-209 as follow:

“Potential limitations, considerations and conclusions.

*Taken together, this study highlights the power of systematically activating BGCs in fungi to identify potential bioactive metabolites. Although our revised strategy—employing a stronger promoter and integration at the yA genomic locus—has proven successful for many BGCs, it does have some limitations. First, eukaryotic gene activation, and SM gene regulation in particular, is often subject to combinatorial control. This regulation requires multiple elements, including global TFs, chromatin-modifying proteins, mediators³³⁻³⁵ and post-translational modifications. Our strategy of single TF OE does not account for this complex coordination and, therefore, could not activate all BGCs, which was nicely demonstrated by the transcriptome analysis in the recent systematic SM TF OE study of *A. niger*¹⁵. A similar transcriptome analysis on our SM TF OE strains would help determine how many of the BGCs analyzed in this study could not be activated. Second, our approach primarily targets TFs located within annotated BGCs. As a result, metabolites produced by BGCs lacking a TF^{36, 37} or by non-clustered biosynthetic genes^{10, 38} may be overlooked. Third, excessive TF expression can lead to off-target activation, resulting in the production of non-specific metabolites and/or altered fungal physiology. While this may complicate the identification of primary BGC-derived metabolites, it could also be advantageous if these non-specific metabolites—typically not produced under normal conditions—exhibit useful bioactivities.*

*While the systematic TF OE strategy has demonstrated efficacy in *A. nidulans*, its broader application to non-model or genetically intractable fungi requires several important considerations. A key prerequisite is the availability of well-characterized genetic tools, including strong inducible promoters, efficient transformation systems, and reliable selection markers—resources often lacking in less-studied fungal species. Moreover, targeted genomic integration, which we achieved at the yA locus, may be inefficient or unpredictable in species with low homologous recombination efficiency or without an effective targeting system (e.g., *nkuAΔ* equivalent strains³⁹). At the transcriptional level, BGCs in some fungi may reside in heterochromatic regions, where silencing mechanisms differ from those in *A. nidulans*⁴⁰, making TF overexpression alone insufficient to overcome repression. Additionally, promoter activity can be context-dependent. Even strong promoters like *xylP* may not function optimally under all conditions⁴¹ or in other fungal species, presumably due to differences in transcriptional machinery or regulatory networks.*

Despite these challenges, advances in synthetic biology and gene editing technologies—such as CRISPR-Cas—have made genome engineering increasingly feasible in fungi. Therefore, the SM TF OE strategy presented in this work offers a powerful approach for uncovering the pharmaceutical potential of diverse fungal species.”

Lines 33-34: it is important to emphasize here that the vast majority of the BGCs identified in fungal genomes are "putative" and lack functional validation. In other words, it is not only that "most remain transcriptionally silent" but also that we lack clarity on how many of them are actually functional.

Response:

We agree with this reviewer that it is important to emphasize the point and have revised the statement as follow: lines 36-41 "*Fungal secondary metabolites represent a rich repertoire of natural products^{1, 2}. Although numerous secondary metabolite (SM) biosynthetic gene clusters (BGCs) have been identified bioinformatically in thousands of fungal genomes, the functionality of these predicted BGCs and their potential to produce metabolites awaits validation. More importantly, most SM BGCs remain transcriptionally silent³ under standard laboratory conditions. Consequently, the vast majority of fungal secondary metabolites have not been identified, and their bioactivity remains unknown.*"

Line 38: a few citations of these low-throughput methods would be helpful here.

Response:

We have added the citations of several research/review papers and revised the sentences (lines 42-44) as "*Despite the development of various methods and significant efforts to explore this rich resource²⁻¹¹, most studies have been limited to a low-throughput approach—activating one cluster at a time to induce the production of its secondary metabolite, which is then tested for bioactivity^{10, 12-14}.*"

2. Keller, N.P. Fungal secondary metabolism: regulation, function and drug discovery. *Nat Rev Microbiol* **17**, 167-180 (2019).
3. Brakhage, A.A. & Schroeckh, V. Fungal secondary metabolites - strategies to activate silent gene clusters. *Fungal Genet Biol* **48**, 15-22 (2011).
4. Brakhage, A.A. Regulation of fungal secondary metabolism. *Nat Rev Microbiol* **11**, 21-32 (2013).
5. Netzker, T. et al. Microbial communication leading to the activation of silent fungal secondary metabolite gene clusters. *Front Microbiol* **6**, 299 (2015).
6. Fischer, J., Schroeckh, V. & Brakhage, A.A. Awakening of fungal secondary metabolite gene clusters. *Gene expression systems in fungi: advancements and applications*, 253-273 (2016).
7. Roux, I. et al. CRISPR-Mediated Activation of Biosynthetic Gene Clusters for Bioactive Molecule Discovery in Filamentous Fungi. *ACS Synth Biol* **9**, 1843-1854 (2020).
8. Chiang, C.-Y., Ohashi, M. & Tang, Y. Deciphering chemical logic of fungal natural product biosynthesis through heterologous expression and genome mining. *Natural product reports* **40**, 89-127 (2023).
9. Woodcraft, C., Chooi, Y.H. & Roux, I. The expanding CRISPR toolbox for natural product discovery and engineering in filamentous fungi. *Nat Prod Rep* **40**, 158-173 (2023).

10. Rabot, C. et al. Transcription Factor Engineering in *Aspergillus nidulans* Leads to the Discovery of an Orsellinaldehyde Derivative Produced via an Unlinked Polyketide Synthase Gene. *J Nat Prod* **87**, 2384-2392 (2024).
11. Wang, M., Chen, L., Zhang, Z. & Wang, Q. Recent advances in genome mining and synthetic biology for discovery and biosynthesis of natural products. *Crit Rev Biotechnol* **45**, 236-256 (2025).
12. Ninomiya, A., Urayama, S.I. & Hagiwara, D. Antibacterial diphenyl ether production induced by co-culture of *Aspergillus nidulans* and *Aspergillus fumigatus*. *Appl Microbiol Biotechnol* **106**, 4169-4185 (2022).
13. Darma, R. et al. Transcriptomics-Driven Discovery of New Meroterpenoid Rhynchospenes Involved in the Virulence of the Barley Pathogen *Rhynchosporium commune*. *ACS Chem Biol* **20**, 421-431 (2025).
14. Shang, Z. et al. Self-Resistance Gene-Guided Discovery of the Molecular Basis for Biosynthesis of the Fatty Acid Synthase Inhibitor Cerulenin. *Angew Chem Int Ed Engl* **64**, e202414941 (2025).

Line 39: given that the authors' approach is only applicable to clusters with an embedded transcription factor gene, pitching this approach as a way "to evaluate the overall pharmaceutical potential" is somewhat of an exaggeration.

Response:

We agree with this reviewer and have removed that statement and replaced it with a statement introducing a similar systematic study of *Aspergillus niger*, highlighting the power of the systematic approach. The revised version (lines 45-48) now reads as “*Notably, a recent study has systematically over-expressed 58 cluster-specific transcription factors (TFs) of Aspergillus niger, leading to the production of potentially novel compounds¹⁵. Although the bioactivity of those compounds was not determined in that study, such systematic approach provides a valuable means to evaluate the pharmaceutical potential of a given fungus.*”.

Lines 84-90: the numbers don't add up here: 38 uncharacterized TFs + 11 known TFs + 4 TFs outside of BGCs based on publications + 7 TFs outside of BGCs based on AspGD information = 60. However, the authors say that "In total, 51 TFs were selected".

Additionally, it would be useful to include this information on Table 1.

Response:

We apologized for the mistake and have corrected the information in the revised manuscript as follows “*This included 33 uncharacterized (Table 1) and 9 known TFs (aflR, afoA, apdR, atnN, dbaA, mdpA, mdpE, pbcR and scpR). Additionally, TF genes outside of the annotated secondary metabolite BGCs but previously associated with SM regulation based on publications (rsmA, napA, mtfA, and hapX) and information from the AspGD database (AN2025, AN2026, AN2241, AN4933 and AN8645) were also included.*”.

As suggested, we have included the information in an extra column in Table 1.

Table 1 - Summary of SM TF over-expression strains.

No.	TF	Cluster	Type	Source	Ref
1	mdpE	Monodictyphenone (mdp) cluster	Zn(II)2Cys6	Published	47
2	AN0153	Monodictyphenone (mdp) cluster	Myb	SMURF	
3	AN0533	pkdA cluster	Zn(II)2Cys6	SMURF	
4	mdpA	Monodictyphenone (mdp) cluster	WHR	Published	47
5	afoA	Asperfuranone (afo) cluster	Zn(II)2Cys6	Published	48
6	AN10294	AN10289 cluster	Zn(II)2Cys6	SMURF	
7	AN10295	AN10289 cluster	bZIP	SMURF	
8	AN10300	AN10297 cluster	Zn(II)2Cys6	SMURF	
9	AN10491	AN10486 cluster	Zn(II)2Cys6	SMURF	
10	AN10600	AN4827 cluster	Zn(II)2Cys6	SMURF	
11	pbcR	AN1594 (PbcR) cluster	Zn(II)2Cys6	Published	49
12	AN1678	AN1680 cluster	Zn(II)2Cys6	SMURF	
13	AN2025		Winged Helix Regulator	AspGD	
14	AN2026		uncertain	AspGD	
15	AN2036	Pkh cluster	Zn(II)2Cys6	SMURF	
16	AN2241		Zn(II)2Cys6	AspGD	
17	AN2553	Emericellamide (eas) cluster	Zn(II)2Cys6	SMURF	
18	AN3269	AN3273 cluster	Zn(II)2Cys6	SMURF	
19	AN3280	AN3273 cluster	Zn(II)2Cys6	SMURF	
20	AN3391	Microperfuranone cluster	C2H2 and Zn(II)2Cys6	SMURF	
21	scpR	inp cluster	C2H2	Published	50
22	AN3501	inp cluster	Zn(II)2Cys6	SMURF	
23	AN3502	inp cluster	Zn(II)2Cys6	SMURF	
24	AN3911	AN10486 cluster	Zn(II)2Cys6	SMURF	
25	rsmA		bZIP	Published	51
26	AN4837	AN4827 cluster	Zn(II)2Cys6	SMURF	
27	AN4933		Zn(II)2Cys6	AspGD	
28	AN5609	AN5610 cluster	Zn(II)2Cys6	SMURF	
29	AN6430	AN6431 cluster	Zn(II)2Cys6	SMURF	

30	AN6788	xptA-containing cluster	Zn(II)2Cys6	SMURF	
31	AN6790	xptA-containing cluster	Zn(II)2Cys6	SMURF	
32	AN7061	pkg cluster	Zn(II)2Cys6	SMURF	
33	AN7072	pkg cluster	Zn(II)2Cys6	SMURF	
34	AN7073	pkg cluster	Zn(II)2Cys6	SMURF	
35	napA		bZIP	Published	52
36	afIR	Sterigmatocystin (stc) cluster	Zn(II)2Cys6	Published	53
37	atnN	AN7884 cluster	Zn(II)2Cys6	Published	54
38	dbaA	DbA and F9775 clusters hybrid 1	Zn(II)2Cys6	Published	30
39	AN7919	DbA and F9775 clusters hybrid 2	Zn(II)2Cys6	SMURF	
40	AN7921	DbA and F9775 clusters hybrid 2	Zn(II)2Cys6	SMURF	
41	AN7923	DbA and F9775 clusters hybrid 2	Zn(II)2Cys6	SMURF	
42	AN8111	AN8105 cluster	Zn(II)2Cys6	SMURF	
43	hapX	AN8249 cluster	bZIP	Published	55
44	AN8391	Austinol (aus) cluster	Zn(II)2Cys6	SMURF	
45	apdR	Aspyridone (asp) cluster	Zn(II)2Cys6	Published	17
46	AN8506	Terriquinone (tdi) cluster	Zn(II)2Cys6	SMURF	
47	AN8509	Terriquinone (tdi) cluster	Zn(II)2Cys6	SMURF	
48	AN8645		Zn(II)2Cys6	AspGD	
49	mtfA		C2H2	Published	56
50	AN8902	AN8910 cluster	Zn(II)2Cys6	SMURF	
51	AN9013	AN9005 cluster	Zn(II)2Cys6	SMURF	

Line 106: "assay" -> "assayed"

Response:

Corrected.

Lines 131-134: The authors have only tested against 2 bacteria and 1 fungus. There is also no information on whether *A. nidulans* does (or does not) naturally interact with these three species. I just don't see how these data remotely justify the hypothesis that there is "a selective pressure for *A. nidulans* to evolve metabolites for competition against bacteria over other fungi". I suggest deletion.

Response:

The statement "*selective pressure for A. nidulans to evolve metabolites for competition against bacteria over other fungi*" has been removed.

The revised version can be found in lines 152-154 and reads as "*The biological significance of these stage-specific activities is currently unclear. Overall, there are less metabolites with anti-fungal activities comparing to anti-bacteria activities.*".

Line 221: please state the bacterial strains tested.

Response:

The information has been added.

Line 273-274: "*To detect the antibacterial activity of EtOAc extracts from each OE strain, the gram-positive bacteria Staphylococcus aureus (ATCC25904) and Bacillus subtilis (ATCC 21336) were used.*"

Line 230: please state the fungal strain tested.

Response:

The information has been added.

Line 283-284: "*To test the antifungal activity of these EtOAc extracts, Candida albicans (SC5314) was used as the test strain.*"